# Can Superhydrophobic PET Surfaces Prevent Bacterial Adhesion?

**DOI:** 10.3390/nano13061117

**Published:** 2023-03-21

**Authors:** Tugce Caykara, Sara Fernandes, Adelaide Braga, Joana Rodrigues, Ligia R. Rodrigues, Carla Joana Silva

**Affiliations:** 1CENTI-Center for Nanotechnology and Smart Materials, Rua Fernando Mesquita 2785, 4760-034 Vila Nova de Famalicão, Portugal; 2CEB-Centre of Biological Engineering, Universidade do Minho, Campus de Gualtar, 4710-057 Braga, Portugal; 3CITEVE-Portuguese Technological Centre for the Textile and Clothing Industries, Rua Fernando Mesquita 2785, 4760-034 Vila Nova de Famalicão, Portugal

**Keywords:** nanoparticles, surface modification, polyethylene terephthalate, superhydrophobic, bacterial adhesion

## Abstract

Prevention of bacterial adhesion is a way to reduce and/or avoid biofilm formation, thus restraining its associated infections. The development of repellent anti-adhesive surfaces, such as superhydrophobic surfaces, can be a strategy to avoid bacterial adhesion. In this study, a polyethylene terephthalate (PET) film was modified by in situ growth of silica nanoparticles (NPs) to create a rough surface. The surface was further modified with fluorinated carbon chains to increase its hydrophobicity. The modified PET surfaces presented a pronounced superhydrophobic character, showing a water contact angle of 156° and a roughness of 104 nm (a considerable increase comparing with the 69° and 4.8 nm obtained for the untreated PET). Scanning Electron Microscopy was used to evaluate the modified surfaces morphology, further confirming its successful modification with nanoparticles. Additionally, a bacterial adhesion assay using an *Escherichia coli* expressing YadA, an adhesive protein from Yersinia so-called Yersinia adhesin A, was used to assess the anti-adhesive potential of the modified PET. Contrarily to what was expected, adhesion of *E. coli* YadA was found to increase on the modified PET surfaces, exhibiting a clear preference for the crevices. This study highlights the role of material micro topography as an important attribute when considering bacterial adhesion.

## 1. Introduction

Bacterial adhesion and subsequent biofilm formation create several risks and challenges in both healthcare and industrial applications [1,2]. Some of the approaches to prevent bacterial adhesion include, for instance, leaching biocides from surfaces which kill both the adhered and nearby bacteria. However, there is a growing concern regarding the increased use of biocides as it can cause increased antimicrobial resistance, therefore its use must be limited [1]. On the other hand, superhydrophobic surfaces have been gaining attention due to their self-cleaning and anti-adhesive properties. Superhydrophobic surfaces are defined as surfaces with a water contact angle higher than 150°. These surfaces combine both low surface energy and nanopatterned structures. Several authors have reported a decreased bacterial adhesion on such surfaces due to the formation of solid-air-liquid interfaces, i.e., air pockets between bacteria and the surface [3] that reduce adhesion by preventing bacteria to approach the surface [1] and provide an easy removal of bacteria as the result of a weaker binding at the interface [4].

Since early 2000s after the discovery of hierarchical micro/nanostructure of the Lotus leaf [5,6] and with the development of the technology, interest in superhydrophobic surfaces has increased and this has also boosted the search for other superhydrophobic surfaces in nature, such as those existing in water strider legs, mosquito eyes, rice leaves, butterfly wings, and gecko feet [7]. Different methodologies including plasma, etching, electrospinning, sol-gel, self-assembly, layer-by-layer, surface initiated radical polymerization, and others can be used to develop superhydrophobic surfaces [7]. Particularly, the sol-gel method offers cheap and versatile fabrication techniques without the need of a special equipment [8] and thus, it has attracted significant attention. This technique includes solution gel transition by hydrolysis and condensation with acidic/basic catalysis [9]. In a study done by Zhang et al. [10], aminopropyltriethoxysilane to form micro/nanohierarchical structures with 1H,1H,2H,2H-perfluorooctyltriethoxysilane for low surface energy was utilised via the Stober method. The silica sol then coated on glass slides with silicone resin which resulted in water contact angle of 168°. The modified surface further showed significantly low adhesion towards *Navicula pelliculosa*.

The most commonly used low surface energy molecules to form superhydrophobic surfaces include fluorocarbon chains such as trifluoropropyl [11], polytetrafluoroethylene [12], and 1H, 1H, 2H, 2H-perfluorodecanethiol [13]. In a study by John et al. [14], nanoimprint lithography with a low surface energy molecule, 1H,1H,2H,2H-perfluorodecyl acrylate was utilised to form superhydrophobic surfaces which performed to confer the surface anti-biofouling properties against *E. coli*. Although fluorocarbon chain was a common choice, superhydrophobic surfaces were also formed with other means. In a study done by Zhao et al. [15], aminated polydimethyl siloxane was used with epoxy resin and was cured with isocyanate crosslinker where a bacterial reduction against *E. coli* at 79% was observed with modified samples. In another study, Han et al. [16] studied silicone dioxide (SiO_2_) and titanium dioxide (TiO_2_) NPs deposited on a surface which was pre-modified with mussels’ adhesive protein, and the modified material showed excellent anti-biofouling properties. Nanoparticles formed via sol-gel method are generally applied as coating on surfaces via solvent evaporation [17], dip coating/padding [18,19], spray coating [20], and others. However, the superhydrophobic surfaces still lack durability and more studies need to be done to improve the longevity of the material properties [1,8]. 

Several studies on the modification of material surfaces with NPs proved to be effective at minimizing the contact between the liquid and the solid surface. However, it is still challenging to create stable and durable superhydrophobic surfaces and most of the methods being used do not include chemical attachment of the NPs to the surface, hence making it less durable [1]. An alternative method to increase its durability can be the in situ growth of NPs on the surface due to presence of covalent bond between nanoparticles and substrate [21]. Polyethylene terephthalate (PET) has been widely used in biomedical studies. Thus, in this work, polyethylene terephthalate surfaces were modified with in situ growth of silica NPs to increase the durability of the coating through an interfacial bonding [9,21], and the obtained surfaces were characterized by contact angle, atomic force microscopy, scanning electron microscopy, and dynamic light scattering.

## 2. Materials and Methods

### 2.1. Materials

Polyethylene terephthalate (PET) film (Melinex 401 CW/100 μm) was acquired from Putz Folien, Taunusstein, Germany. Dynasylan F 8261 (FAS13: triethoxy(3,3,4,4,5,5,6,6,7,7,8,8,8-tridecafluoro-1-octyl) silane) from Evonik, Germany, was purchased from Safic-Alcan, Milheirós, Portugal. Nitric acid (70%) was from ChemLab, Zedelgem, Belgium; ammonia (30%) was from Labkem, Spain; tetraethyl ortosilicate (TEOS, 98%) and diodomethane (99%) were purchased from Sigma-Aldrich, St. Louis, MO, USA; hydrochloric acid (37%) was from VWR Chemicals, Fontenay-sous-Bois, France; ethyl alcohol (99.8%) from Aga, São Brás de Alportel, Portugal; sodium hydroxide from Eka, Portugal; sodium chloride and potassium phosphate monobasic were from Fisher Chemicals, Loughborough, UK; potassium chloride from Panreac, Barcelona, Spain; sodium phosphate was from Merck, Darmstadt, Germany.

### 2.2. Pre-Treatment of the Surfaces

In order to clean the surfaces prior to functionalization, 3 × 6 cm sized PET films were immersed in ethyl alcohol for 15 min. Afterwards, the PET films were rinsed with ethyl alcohol and left to dry at room temperature. For the hydrolysis of PET surface, the alkali method previously reported [22,23] was used. Briefly, PET films were immersed in NaOH (4 M) at 70 °C under constant shaking at 100 rpm for 2.5 h and then were washed with HCl (1 M) until neutral pH. 

### 2.3. In Situ Nanoparticle (NP) Modification of PET Surfaces

PET films were immersed in a solution of 332 mL of ethanol, 90 mL of ultrapure water, and 100 μL nitric acid (as provided, 70%) after being heated to 75 °C. Soon after the PET films were immersed, 5 mL TEOS was poured into the solution under stirring. After 7 min, 32 mL of ammonia and 20 mL TEOS were added with a flow rate of 30 mL/min. After 2.5 h under constant stirring, 3 mL of FAS13 was added to the solution and left to react for 2.5 h. After that time, the sample was removed, rinsed with ethanol, and ultrasonicated for 10 min in ethanol. Finally, the sample was rinsed with distilled water and put in the oven to dry for 30 min at 80 °C. This experimental procedure for the in situ NPs formation was adapted from Chen et al. [24]. 

### 2.4. Contact Angle

Static contact angle measurements using the Optical Tensiometer Attension Model Theta Basic by Biolin Scientific were performed at room temperature. For the water contact angle (WCA) measurement, MiliQ water with a droplet size of 3 µL was used. The measurements were repeated at least four times for each sample at different surface locations. Contact angle measurements were also performed with diiodomethane to calculate the polar and disperse components of surface energy by Owens-Wendt-Rabel & Kaelble Model (OWRK) equation [25].

### 2.5. Atomic Force Microscopy (AFM)

Atomic Force Microscopy (AFM) measurements were performed by a Keysight Technologies 5500 Atomic Force Microscope. The tapping mode (Mac Mode) in air at room temperature was used. The cantilever material was silicone with a spring constant of 13–77 N/m and radius < 10 nm. Keysight PicoView and Gwyddion software were used to analyse the images, with a surface area of 5 × 5 μm.

### 2.6. Scanning Electron Microscopy (SEM)/Energy Dispersive Spectroscopy (EDS)

The morphological and chemical analyses of the samples were performed at the Materials Characterization Services Laboratory of the University of Minho (SEMAT-UM), using a Scanning Electron Microscopy (SEM) NanoSEM–FEI Nova 200 (FEG/SEM). Before the morphological analysis, the samples were coated with a thin film of Au/Pd (80/20 by weight) 10 nm thick. For the morphological analysis, a secondary electron detector with 10 KeV energy and a working distance between 7 and 8 mm, were used. For the chemical analysis, energy dispersive spectroscopy (EDS) technique was performed by an Integrated System EDAX–Pegasus X4M, using 15 KeV energy, at a working distance of approximately 6.5 mm.

### 2.7. Dynamic Light Scattering (DLS)

The size and zeta potential of functionalized NPs were characterized by Dynamic Light Scattering (DLS) using a ZETASIZER NANO-ZS90 (Malvern Paralab) with appropriate software. Dispersions of 3 mg of NPs in 10 mL of ethanol were prepared at room temperature by collecting NPs from three different batches with three replications. 

### 2.8. Bacterial Adhesion

Bacterial adhesion tests were performed according to the methodology previously described [26]. Briefly, 1 × 1.5 cm sized samples were first cleaned by immersing in 1% detergent solution in ultrasonic bath. The samples were immersed in 1× PBS (phosphate-buffered saline, pH 7.4: 137 mM NaCl, 2.7 mM KCl, 10 mM Na_2_HPO_4_ and 2 mM KH_2_PO_4_) solution with *E. coli* expressing YadA [26,27] with an OD_600 nm_ of 0.8 for 1 h, 4 h, and 6 h. Afterwards, PET samples were removed and washed gently with 1× PBS buffer to remove the unattached bacteria. A fluorescence microscope (Olympus BX51), at 60× magnification, coupled with an DP71 digital camera were used to visualise the adhered bacteria with the fluorescence imaging at 470–490 nm excitation and a 520 nm filter in the microscope optical path. 

## 3. Results and Discussion

### 3.1. Surface Modification and Reaction Mechanism

The modification of polymeric surfaces starts with an initial pre-treatment since most polymer surfaces have good chemical resistance [28]. Like other polymers, PET surface is also quite inert. Nevertheless, different techniques can be applied to introduce functional groups onto the PET surface. Hydrolysis is one of the functionalization methods used [29] to generate hydroxyl groups at the surface [22] for further reaction with the silanol groups present, for instance, on the herein prepared NPs [30]. Furthermore, reaction of organosilanes can be obtained in both acidic and basic reaction conditions, where silanes first go through hydrolysis to form silanol groups and then a covalent bond is established with hydroxyl groups on the surface of the substrate by condensation [30]. In acidic solutions, the rate of hydrolysis increases causing stable silanol groups in the solution, whereas basic conditions can catalyse both hydrolysis and self-condensation [31]. Therefore, acidic conditions were first used to allow the reaction of silanes with the PET surface, followed by an exposure to basic conditions where polycondensation and NPs formation on the surface occurred [24]. Finally, a fluoroalkylsilane was added to the solution in order to functionalize the formed nanoparticles and obtain a surface with low surface energy. The proposed reaction mechanism is illustrated in Figure 1. 

### 3.2. Contact Angle and Surface Energy

The surface’s hydrophobicity of a material can be increased using low surface energy molecules including methylated and fluorinated carbons. These groups can be ordered by decreasing surface energy as follows: -CH_2_ > CH_3_ > CF_2_- > CF_2_H > CF_3_. However, the surface modification with the lowest surface energy molecules led to WCA of 119° on a smooth surface [1]. Therefore, it is not possible to create superhydrophobic surfaces using only low surface energy molecules and some surface roughness is also required. Thus, the fluorinated NPs treated surfaces prepared in this work comprise a promising approach that can be used to achieve superhydrophobic surfaces. 

The change in the surface energy due to the chemical modification was assessed by measuring contact angle of samples with water and diiodomethane. As it was shown in Figure 2, the obtained results showed that untreated PET surfaces exhibited a WCA of 69 ± 10°, while hydrolysed surfaces possessed a WCA of 62 ± 6° which agreed with other studies [22,23]. For the NPs treated surfaces (PET NP), the measured WCA was 156 ± 12°. In addition, the WCA for this surface was maintained after ageing tests, as the WCA was found to be 157 ± 10° after 2 weeks and 151 ± 14° after 3 weeks of storage. Furthermore, the samples were washed with water and ethanol, where each cycle of washing consisted of immersing the sample in water for 30 min at 40 °C and 100 rpm, to evaluate the durability of the treatment under these conditions. After 10 washing cycles with water, the WCA was 160 ± 9°, and after 10 additional cycles of washing with ethanol, the WCA was 157 ± 10°, thus showing that the functionalized PET NP samples possessed a resistant superhydrophobic character under the tested conditions. Furthermore, material surface free energy (SFE) values can also influence bacterial adhesion. A low spreading of bacteria was found when the polar component of surface energy was lower than 5 mJ/m^2^ [1] and the biomaterial had a low surface energy [3]. Table 1 shows that the total SFE along with its polar component has increased from 49 mJ/m^2^ and 7 mJ/m^2^ for untreated PET to around 53 mJ/m^2^ and 11 mJ/m^2^ for hydrolysed PET. The total SFE and the polar component of SFE were further decreased to around 7 mJ/m^2^ and 3 mJ/m^2^, respectively, with the NP modification. 

### 3.3. Atomic Force Microscopy (AFM) and Surface Roughness

It is well known that surface roughness is necessary to achieve superhydrophobic surfaces [1]. In Table 2, surface roughness values for untreated and treated PET determined by AFM technique are summarized. Untreated PET surfaces presented a smoother surface topography with Ra value of 4.8 ± 1.0 nm when compared to hydrolysed PET (27.8 ± 2.0 nm) and to NPs treated PET (104 ± 20 nm). This last one was considerably rougher, being this an important aspect for the transitioning of the surfaces to the superhydrophobic state [32].

When the surfaces were visualized by AFM, the topography of hydrolysed PET was found to be an irregular granular structure with pits on the film surface, whereas the NPs modified PET presented the typical appearance of NP like structures (Figure 3). Interestingly, the hydrolysed surfaces also present small granular like structures on the surface intercalated with pits which are created from the etching process, and these can be better identified by SEM images (Figure 4c). Therefore, the topography of PET surfaces was further studied by SEM analysis to confirm the presence of these structures.

### 3.4. Scanning Electron Microscopy (SEM)/Energy Dispersive Spectroscopy (EDS) and Dynamic Light Scattering (DLS) Measurements

SEM/EDS analysis was performed to characterize the surfaces morphology and chemistry. The morphological changes on both hydrolysed and NPs modified samples at different magnifications can be seen in Figure 4. Hydrolysed PET shows signs of etching, which was expected due to breakage of ester groups on the surface of the PET film into hydroxyl and carboxyl groups [22]. It is well known that etching of PET surface depends on the material characteristics such as crystallinity and thickness. During hydrolysis, amorphous regions erode faster than crystalline regions leaving a non-uniform surface structure after short hydrolysing times [23]. The formed hydroxyl groups on the PET surface are the probable location where the covalent bonds with TEOS will be established, and that will lead to the subsequent formation of the NPs on the surface, which can be seen for instance in Figure 4f.

Furthermore, the cross-section of PET NP (Figure 5) shows that polycondensation resulted in a clear layer of silica NPs with a thickness around 711 nm on the top of the PET substrate. The growth of the layer thickness is probably due to continuous polycondensation on the surface. In Figure 5, the NPs on the surface presented a heterogeneous dimension, with values that vary from 326 nm to 151 nm and with many smaller ones.

The NPs formed during the surface modification process and that remained in the solution, were also analysed by DLS to determine their size and zeta potential. The average NP size in the solution was 366 ± 26 nm with a polydispersity index (PdI) of 0.27 ± 0.13. PdI is used to describe the particle size distribution [33] and a value lower than 0.1 is indicative of uniformity [33]. While a PdI between 0.03 and 0.06 indicates a monodisperse solution [34], a PdI higher than 0.5 suggests a wide range of particle size distribution [33]. The differences in the NPs size between the surface and the solution herein observed (Figure 5) might be due to growth of the layer on the surface and continuous polycondensation which is embedding some nanoparticles while allowing the formation of new ones. Moreover, the DLS analysis provided a zeta potential of −34 ± 2 mV, indicating a stable NPs solution since a zeta potential around −30 mV can be considered a moderately stable solution [35].

In Figure 6 the PET surface morphology after 10 cycles of washing with water followed by 10 cycles of washing with ethanol can be observed. Although at lower magnification, washed and unwashed samples seem to have similar features, at higher magnifications it is possible to visualize a NPs free area in the NPs layer on the PET surface (Figure 6a) which indicates that parts of the coating layer were removed from the surface due to washing. Additionally, by observing the sample cross-section, it can be observed that the layer thickness is lower after washing (Figure 6b), which can be explained by the loss of the outermost NPs of the coating layer. It is also important to note that the sizes of the NPs seem to be smaller after washing. Although covalent bonding of the NPs coating layer to the PET surface was expected, it is possible that some NPs were only adhered or were weakly bonded to the surface, or even were only trapped in the network of bonds formed during polycondensation. Nevertheless, the removal of these fluoroalkylsilane functionalized NPs layer suggests that the treated surfaces have limited durability, even though they continue to provide superhydrophobic surface properties after the washing cycles performed with water and ethanol, as previously shown with WCA measurements. These results strengthen the importance of conducting additional testing such as abrasion when determining the durability of these surface treatments.

The treated and untreated samples were further analysed by EDS to determine the chemical composition of the surfaces (Table 3). The elemental analysis of samples showed that untreated and hydrolysed surfaces were mainly constituted by carbon and oxygen atoms, while the NPs treated surfaces, including washed samples, also possessed silicon and fluorine in their composition, thus confirming the modification of the surfaces with TEOS and FAS13.

### 3.5. Bacterial Adhesion Tests

Finally, the performance of the developed modified PET films was evaluated regarding their bacterial anti-adhesion properties. For this purpose, a heterologous *E. coli* expressing the adhesin YadA was used similarly to previous reports [26]. This protein was cloned into a plasmid containing a reporter protein (GFP) to enable its visualization through fluorescence microscopy. Therefore, it was possible to observe the cells binding to the tested materials. The anti-adhesive mechanism of superhydrophobic surfaces depends on the formation of air pockets on the surface that impair the contact between the bacteria and the surfaces. However, the entrapped air from the air pockets will eventually be replaced over time by the immersion of the solution, potentially leading to an increased adhesion [1,36]. Thus, the duration of the experimental setup to assess adhesion can be an important parameter. For this reason, the untreated and fluorinated-NPs treated samples were incubated with the bacterial solution for 1 h, 4 h, and 6 h. Figure 7 illustrates the fluorescence microscopy images captured for the different adhesion time periods.

From Figure 7, it seems that after incubation with bacteria for different time periods, the modified PET samples do not seem to show anti-adhesive properties, when compared to the untreated materials using the same incubation times. This can be further confirmed in the results presented in Figure 8, since the differences between the number of bacteria present in untreated and treated samples were not statistically significant (*p* < 0.05). However, this result differs from other studies that reported a decreased bacterial adhesion when testing other superhydrophobic materials [37,38,39]. In our study, the PET modified materials were found to promote the bacterial adhesion over time. In order to better understand these results, the samples immersed in bacterial solution for 4 h were further characterized by SEM, where adhered bacteria on top of the treated surface can be observed (Figure 9). In Figure 9b, bacteria seem to position itself in the irregularities where the surface area available for cell adhesion is high. This kind of behaviour has also been reported about other superhydrophobic materials where surface irregularities were also found to play a role in promoting bacterial adhesion [40]. In addition, the bacterial cell structure is also known to affect the adhesion of *E. coli* that holds a thinner peptidoglycan layer, making the bacterial cell more flexible when adhering on rough surfaces [41]. Besides, in this study, the *E. coli* is expressing the YadA adhesin which can be helpful when maturing the adhesion [42]. Furthermore, *E. coli* can explore the local topography through flagella which can improve its adhesion on surfaces with microtopographies [43,44]. Moreover, although it was not investigated in this study, it is possible that the surface wetting transition from Cassie-Baxter to Wenzel state might have been displacing air pockets and causing an increased adhesion [45]. Thus, altogether, these factors that confer the bacterial cell the ability to fit irregularities in the modified surface can explain the increased adhesion that was herein observed. Additionally, microtopography and nanotopograhy can be considered crucial factors at governing bacterial adhesion in this study.

## 4. Conclusions

Superhydrophobic materials have been shown to possess bacterial repellent properties by several authors. In situ growth of NPs on the PET surfaces was studied to create a durable superhydrophobic surface. The WCA of treated surfaces showed a significant increase (226%) to a value of 156 ± 12°, when compared to untreated PET (69 ± 10°). Surface roughness increased to 104 ± 21 nm from the initial 5 ± 1 nm value. This surface modification was further characterised by SEM/EDS showing that the surfaces were properly modified with NPs. Although the samples washing with water and ethanol removed some of the bounded NPs from the material surface, the WCA was still high after the performed washing cycles with water (160 ± 9°) and additional washing cycles with ethanol (157 ± 10°), hence indicating that the superhydrophobicity of the surface was maintained in the tested conditions. These results demonstrated that the functionalized surfaces have some degree of durability. Furthermore, bacterial adhesion onto the treated surfaces was evaluated and it was shown that these surfaces promoted bacterial adhesion, contrarily to other reports. This could be explained by the increased bacterial contact within the surface microtopographies, suggesting that surface topography and roughness are crucial factors at determining the bacterial adhesion outcome of a given material.

## Figures and Tables

**Figure 1 nanomaterials-13-01117-f001:**
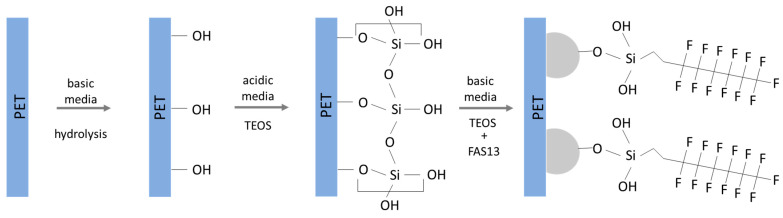
Reaction mechanism proposed for the in situ growth of fluorinated silica NPs on hydrolysed PET.

**Figure 2 nanomaterials-13-01117-f002:**
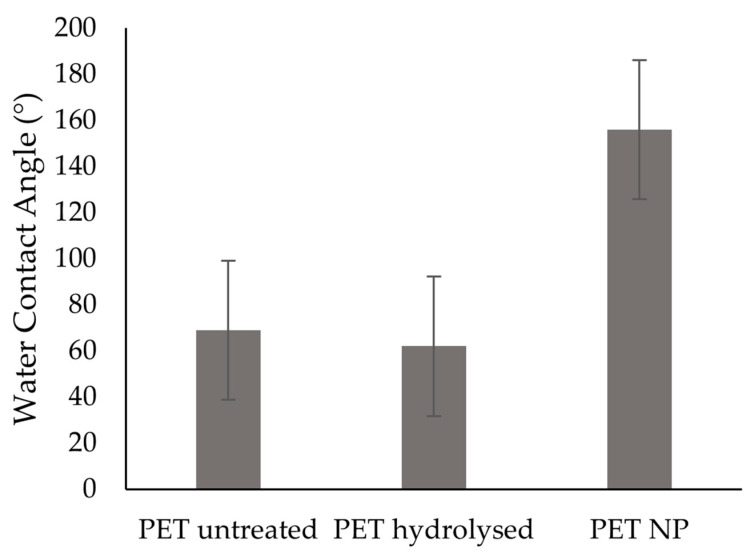
WCA for untreated and treated PET surfaces.

**Figure 3 nanomaterials-13-01117-f003:**
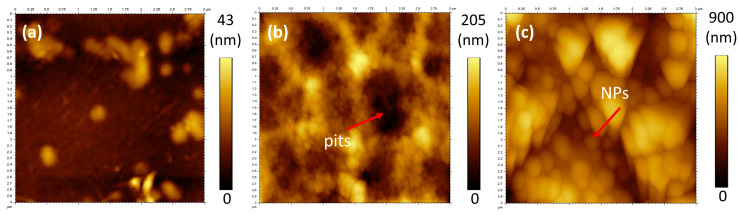
AFM images of PET surfaces—(**a**) PET untreated, (**b**) PET hydrolysed, (**c**) PET NP.

**Figure 4 nanomaterials-13-01117-f004:**
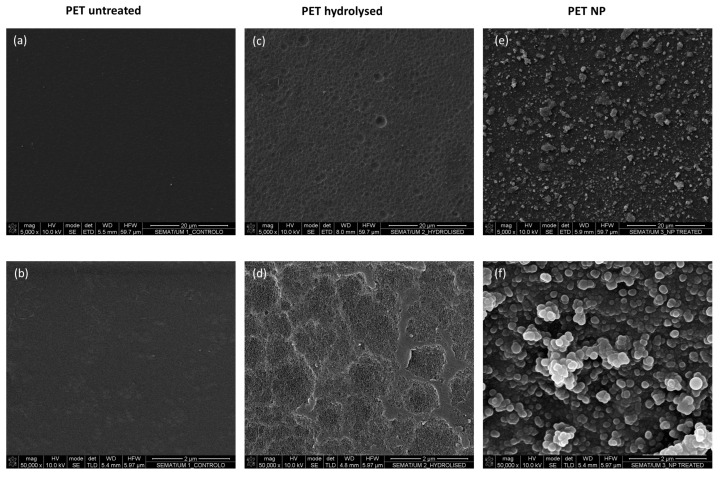
SEM images of untreated (**a**,**b**), hydrolysed (**c**,**d**) and NP modified (**e**,**f**) PET samples, at different magnifications (5000× and 50,000×, respectively).

**Figure 5 nanomaterials-13-01117-f005:**
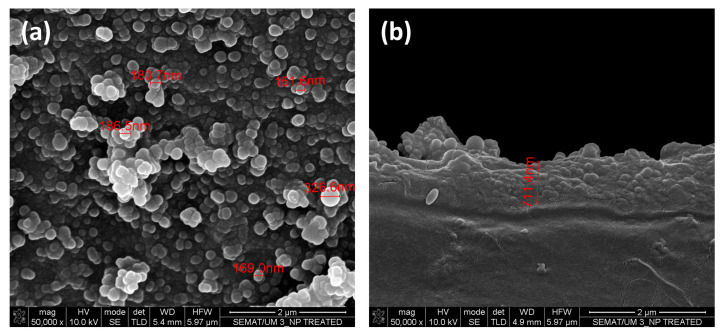
SEM images of PET NP modified surface (**a**) top view (**b**) cross-section view at a magnification of 50,000×.

**Figure 6 nanomaterials-13-01117-f006:**
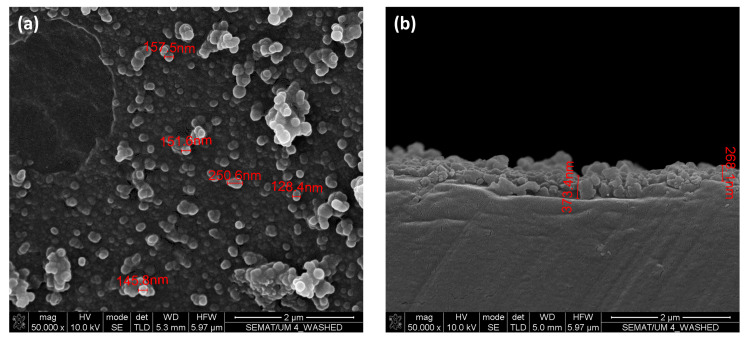
SEM images of PET NP modified surface after 10 cycles of water and ethanol washing: (**a**) top view—50,000× and (**b**) cross-section view—50,000×.

**Figure 7 nanomaterials-13-01117-f007:**
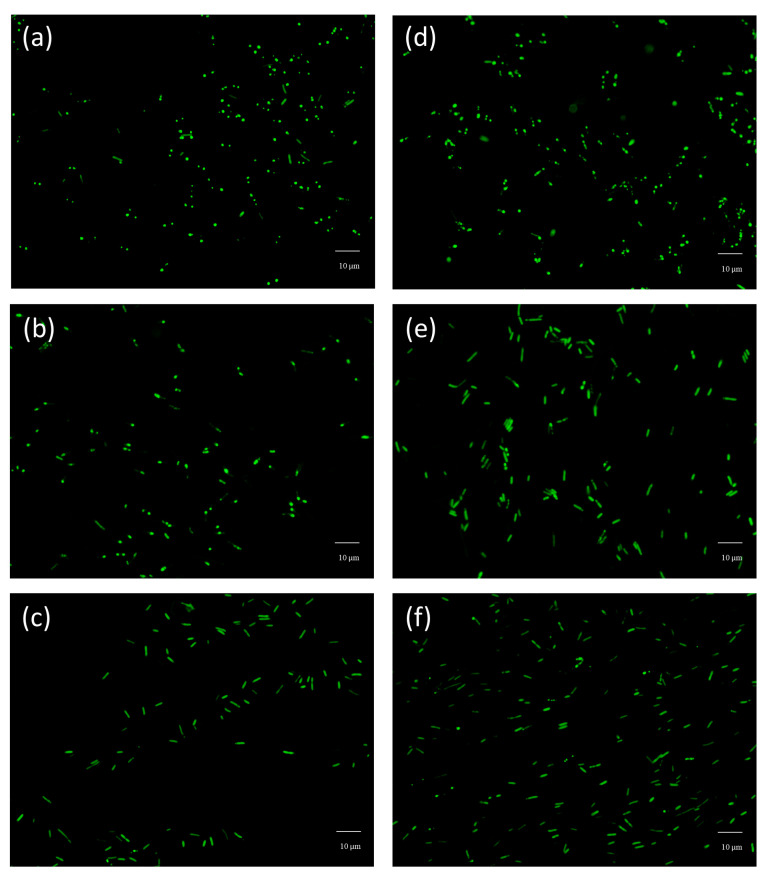
Fluorescence microscopy images of PET samples after incubation with bacteria: (**a**) PET untreated after 1 h; (**d**) PET NP after 1 h; (**b**) untreated PET after 4 h; (**e**) PET NP after 4 h; (**c**) untreated PET after 6 h; (**f**) PET NP after 6 h. Scale bar, 10 µm.

**Figure 8 nanomaterials-13-01117-f008:**
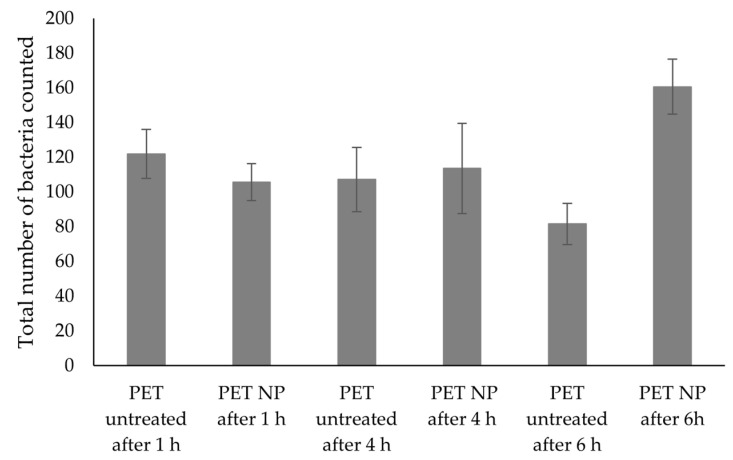
Number of bacteria attached to untreated and treated PET samples, after incubation for 1, 4 and 6 h. Data are presented as means ± SD (n = 6).

**Figure 9 nanomaterials-13-01117-f009:**
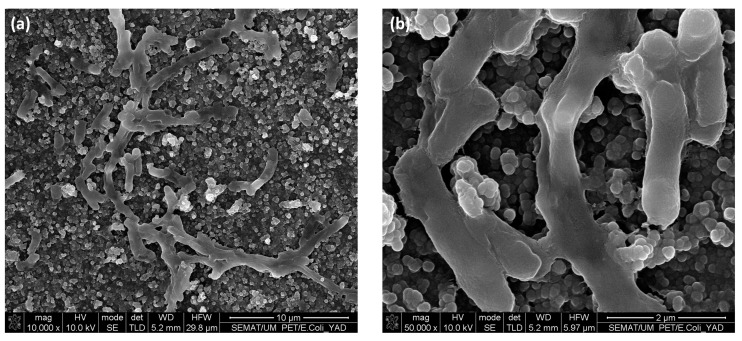
SEM visualization of NPs treated samples with different magnifications (**a**) 10,000× and (**b**) 50,000×, after the bacterial adhesion test (4 h).

**Table 1 nanomaterials-13-01117-t001:** Contact angle values determined with water and diiodomethane, and surface free energy (SFE) of PET untreated, PET Hydrolysed and PET NP.

Samples	Water Contact Angle (°)	Diiodomethane Contact Angle (°)	SFE Disperse mJ/m^2^	SFE Polar mJ/m^2^	SFE Total mJ/m^2^
**PET untreated**	69 ± 10	35 ± 4	42.0 ± 1.8	7.2 ± 4.0	49.2 ± 4.2
**PET hydrolysed**	62 ± 6	35 ± 3	41.9 ± 1.5	10.8 ± 3.1	52.7 ± 3.7
**PET NP**	156 ± 12	114 ± 14	1.3 ± 1.2	3.3 ± 5.7	6.7 ± 4.3

**Table 2 nanomaterials-13-01117-t002:** Surface roughness values (Ra) of treated and untreated PET samples determined by AFM.

Samples	Roughness, Ra (nm)
**PET untreated**	5 ± 1
**PET hydrolysed**	28 ± 2
**PET NP**	104 ± 20

**Table 3 nanomaterials-13-01117-t003:** SEM-EDS results for atomic concentrations of untreated and treated PET samples (n.d.: not detected).

Samples/Elements (% Atomic)	C	O	Si	F
**PET untreated**	86	14	n.d.	n.d.
**PET hydrolysed**	85	15	n.d.	n.d.
**PET NP**	62	28	8	2
**PET NP after 10 washing cycles with water and ethanol**	62	27	7	4

## Data Availability

The data presented in this study are available on request from the corresponding author. The data are not publicly available due to institutional internal regulations.

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
