# Peer review of "Can Superhydrophobic PET Surfaces Prevent Bacterial Adhesion?"

_nanomaterials, 2023, doi:10.3390/nano13061117_

Round 1

Reviewer 1 Report

1. The reaction of in-situ growth of fluorinated silica NPs on hydrolysed PET should be analyzed quantitatively using various equipment to define the growth of functional groups 

2. Figures of water contact angle (in Table 1) should be presented as a separate Figure

3. In Figure 6. Live/dead cell staining of bacteria is required.

4. Quantitative analysis of bacterial growth on the PET (treated/untreated) should be presented in a separate figure.

5. In Figure 7. Bacterial adhesion photographs of untretaed PET and treated PET (hydrolyzed) should be also presented and compared.

Reviewer 2 Report

For the comments - see the PDF file with color highlights.

Reviewer 3 Report

I think that the manuscript is generally well written and presents the results of research well, but unfortunately it needs a few minor corrections. After that, the manuscript will be suitable for publication in the Nanomaterials.

Reviewer 4 Report

Key findings: The author investigated the effectiveness of superhydrophobic PET surfaces in preventing bacterial adhesion.  The author covalently bound SiO2 nanoparticles with fluorinated chains to induce PET surfaces with superhydrophobic characters with the objective to preventing bacterial adhesion.  However, due to the uncontrolled growth of SiO2 NPs on the PET surfaces, the disordered NPs film formed many crevices which leads to the adhesion of bacterial.  This paper showed that in addition to surface contact angle in determining superhydrophobicity, the ordering and assemblies of these SiO2 NPs on the PET surfaces is also important factor in in preventing bacterial adhesion.

Comments and suggestion.

Since surface irregularities has been reported to play a promoting in bacterial adhesion [REF 41], the author should elaborate on the key findings of this paper that would be of interest to the scientific community.  And PET functionalization has also been reported elsewhere.  Can the author elaborate on the major finding or development of this paper?  Without a clear statement, it seems like the author simply reports on an undesired experimental result (non-uniform SiO2 NPs film), which led to an undesired result (bacterial adhesion).  
